# Effects of Forced-Air Precooling on Postharvest Physiological and Storage Quality of Winged Beans

Ying-Che Lee [1], Min-Chi Hsu [2], Jia-Zhu Liao [1], Zhao-Wei Wei [1], Hsin-Ying Chung [1] and Yu-Shen Liang [1,*]

[1] Department of Plant Industry, National Pingtung University of Science and Technology, Pingtung 912, Taiwan
[2] Taiwan Agricultural Research Institute, Council of Agriculture, Taipei 413008, Taiwan
* Correspondence: justinliang@g4e.npust.edu.tw; Tel.: +886-8-7703202#6247

**Abstract:** Winged beans accumulate abundant field heat following harvest, and their shelf life is shortened if precooling is not performed promptly. In the present study, top-suction forced-air precooling (FC) was employed to rapidly remove field heat from pods, and its effects on winged bean pod storage quality and shelf life were assessed. After postharvest FC to remove field heat from winged bean pods, the mean 1/2 precooling time was 5.8 min and 7/8 precooling time was 14.7 min, which was 9.6 times and 11.7 times faster than room cooling (RC), respectively. Moreover, after FC was applied to remove field heat, the weight loss rate at 7/8 precooling time was 0.92%, significantly lower than that after RC was applied (1.98%). FC could delay decay, and the decay rate was only 18% on day 14 storage, which was lower than 52% of RC. During 12 °C and 85% relative humidity (RH) storage, the shelf life of winged bean pods in the FC group was 14.8 days, which was significantly longer than that of the pods in the RC group (10.6 days). In conclusion, FC is an effective precooling method to rapidly remove field heat postharvest and maintain the storage quality of winged beans.

**Keywords:** 1/2 precooling time; respiration rate; decay rate; storage life

## 1. Introduction

Winged bean (*Psophocarpus tetragonolobus* (L.) DC) is also known as four-angled bean, Goa bean, or dragon bean. Winged bean is a dicotyledonous plant of the Faboideae subfamily of the Fabaceae family [1]. Winged bean has a closed flower system, is self-pollinated, and does not require insects for pollination [2]. Winged bean is the most common legume in Southeast Asia and Papua New Guinea, although its commercial production scale remains small, and the cultivation area is limited [3]. The winged bean is a perennial climber, and its pods are significantly different from common beans, with a characteristic winged or star shape. The pods differ in terms of length, width, and color, although the typical colors are green or purple [4]. Many parts of winged beans are edible, including immature pods, leaves, and tubers. In Taiwan, immature pods are commonly used as vegetables. At 2–3 weeks after flowering, the pods are harvested when they are tender and contain less fibers. Winged beans have many health benefits. Fresh winged bean pods contain abundant vitamin C and other minerals and vitamins, such as iron, copper, manganese, and calcium. In addition, thiamin pyridoxine, niacin, and riboflavin are essential B complex vitamins in winged bean [5].

Pod crops are similar to many horticultural crops, as they are prone to decay and rapid decline in quality after harvesting, leading to substantial postharvest losses [6]. In particular, freshly harvested winged bean pods are prone to decay due to their high water content. Therefore, it is recommended that the beans be sold within 24 h postharvest [7]. After harvesting, the pods were stored in high humidity conditions, causing microbial spoilage, or stored at inappropriate low temperatures, causing browning at the tip or edge of the pods and wilting due to the accelerated water loss of the pods due to the impact of chilling injury [7]. After harvesting, fresh immature pods are cooled and stored at 10 °C and

90% RH to reduce water loss and decay. The approximate shelf life is 4 weeks at 10 °C [8]. The respiration rate of winged bean pods was moderate in horticultural crops, similar to the respiration of green snap beans, about 11–20 mL $CO_2$ $Kg^{-1}$. $h^{-1}$ at 5 °C [9]. Silva et al. [10] collected pods 10 days after flowering and used passive modified atmosphere packaging; the pods were stored at 15 °C in cold storage, which extended the shelf life to 24 days.

Precooling extends the shelf life of products through the entire process, from field harvesting to marketing, while reducing the need for urgent consumption by consumers after purchase or critical product processing. This increases consumer satisfaction and promotes purchase [11]. Young pods are tender organs with high respiration rates, which accumulate abundant field heat. This may result in rapid withering and decay after harvesting. According to EO et al. [12], fresh pods must be cooled within 2 h postharvest to maintain their commercial quality. Therefore, rapid postharvest precooling is essential. Currently, pods are mainly precooled through room cooling (RC) after harvesting. Forced-air precooling (FC) is based on RC but involves using an exhaust fan in cold storage to create a pressure difference between two sides of ventilation pores. Due to this, cold air enters one ventilation pore and directly contacts the product while leaving the other pore with field heat, thereby allowing rapid cooling of the product [13]. In addition, the packaging box used does not require waterproofing for ice or crushed ice precooling [14]. FC's cooling speed is faster than RC [15], and it has often been used for precooling fruits, vegetables, or cut flowers [16].

Cortbaoui et al. [17] explored the effects of FC at a wind speed of 3 $L \cdot s^{-1} \cdot kg^{-1}$ and vertical stacking in sweet corn. Compared with RC, the 1/2 precooling time of FC was only 47 min. Moreover, after FC, the high total soluble solids and water content and good fruit appearance were maintained. Finally, the shelf life of fruits was extended by 21 days. In another study, mushrooms were cooled using RC, ice water, and FC. High quality could be maintained during storage after FC, and FC was significantly superior to RC and hydrocooling [18].

Furthermore, FC was combined with modified atmosphere packaging to preserve mushrooms; specifically, FC was applied to rapidly remove the field and respiratory heat in mushrooms, decrease the evapotranspiration and respiration rate and prevent condensation in the packaging. After FC, the weight loss rate of mushrooms was <5%. Simultaneously, mushrooms were stored at 2 °C to maintain good appearance and quality and effectively extend their shelf life [19].

Taitung is a winged bean cultivation area in Taiwan. As winged bean pods harvested during spring and summer accumulate abundant field heat, the temperature of the postharvest pods is around 25–30 °C. Moreover, the shelf life of the pods is shortened if precooling is not conducted rapidly. Therefore, the present study employed FC to rapidly remove field heat from pods and examined its effects on pod storage quality to determine its feasibility.

## 2. Materials and Methods

### 2.1. Materials

Pods of 'Taitung No. 1' winged beans cultivated in Taitung were collected. The pods were harvested for experiments in November 2021 at 20–22 days after flowering. The pods used in this experiment were about 17–20 cm in length and 15–18 g in weight. Within 1 h after harvesting, the beans were transported to the packaging site for postharvest processing. The FC machine used a top suction design. The size of the plastic basket is 62 cm (length) × 43 cm (width) × 19 cm (height). Each plastic basket contains 4.0–4.5 kg of pods. The bottom ventilation hole at the basket's bottom was 0.64 $cm^2$, and the vent hole was 37%. Around 16–20 kg of beans could be processed each time. The inlet wind speed was 1.4 $m \cdot s^{-1}$, the outlet wind speed was 10.8 $m \cdot s^{-1}$, and the mean static pressure in the FC basket was −0.7 hPa. The mean temperature of the precooling warehouse was 8 °C, and RH was 88%. The temperature of the storage warehouse was 12 °C, and the RH was 85%.

### 2.2. Methods

After winged bean pods were harvested in the field, they were immediately transported to the packaging site. After screening, the pods were placed in plastic baskets. Each basket was filled with 4 kg pods. The baskets were directly placed in the precooling warehouse for RC. When the temperature was 7/8 of the precooling time, the pods were packaged in polyethylene bags in the packaging zone in a low-temperature (12 °C) environment. Specifically, $150 \pm 10$ g winged bean pods (around 10–11 pods) were packed in each bag, and the bags were placed in cold storage. For FC, the harvested pods were placed in plastic baskets and directly transferred to the precooling warehouse for FC. Each plastic basket was packed with 4 kg pods, and a top-suction forced-air precooling machine was used to process four baskets each time. Triplicates were set up for every treatment. A thermocouple temperature recorder (HOBO, UX120) was used. A thin (2 mm) thermocouple temperature sensor was inserted in the middle of the pod to record the temperature during precooling. The temperature recorder records once every 30 s. Simultaneously, a temperature recorder (HOBO, U23-001) was used to record the changes in temperature and RH in the precooling warehouse. Precooling was stopped for RC and FC when the temperature at the center of the pod reached 7/8 cooling time. Winged beans for which RC and FC were completed were packaged. Thereafter, the pods were packed into plastic baskets and stored at 12 °C for 14 days in cold storage for relevant quality and shelf life investigations. During precooling, temperature changes at the center of the pod were presented as dimensionless temperature, and 1/2 and 7/8 precooling times were recorded. The dimensionless temperature was calculated as follows:

$$\text{Dimensionless temperature} = \frac{(T\alpha - Ti)}{(T1 - Ti)},$$

where $Ti$ is the temperature of the precooling medium, $T1$ is the initial temperature of the product, and $T\alpha$ is the temperature change of the product during precooling [19].

### 2.3. Respiration Rate and Ethylene Production

First, 100 g winged beans were placed in 0.7 L sample tanks before sealing. A flowing system was used to investigate the respiration rate and ethylene production. The sample tank contained an inlet and outlet, and hourly air intake was 1 L. Air entering the sample tank was air from the outdoor environment that passed through a potassium permanganate-containing ethylene absorbent and deionized water for filtration. Finally, dry and moist air was delivered into the sample tank. A 1 mL syringe was placed in the sample tank outlet to measure the respiration rate for sampling before measurement. The thermal conductivity detector of a gas chromatograph (Shimadzu, GC-8A) and a silica gel column (3 mm × 3 m) were used. The injection port temperature was 100 °C, and the column temperature was 90 °C. The respiration rate was expressed as mg $CO_2$ $kg^{-1}$ $h^{-1}$. A 1 mL syringe was placed in the sample tank outlet to measure the ethylene yield for sampling before measurement. The flame ionization detector of a gas chromatograph (Shimadzu, GC-8A) and a Porapak T 80/100 mesh column (3 mm × 2 m) were used. The injection port temperature was 100 °C, and the column temperature was 80 °C. The ethylene yield was expressed as µL $C_2H_4$ $kg^{-1}$ $h^{-1}$. The influence of temperature on respiration rate was first quantified with the $Q_{10}^R$ (temperature quotient for respiration) values, which is the respiration rate increase for a 10 °C rise in temperature. It can be expressed as:

$$Q_{10}^R = \left(\frac{R2}{R1}\right)^{10/T2-T1}$$

where $R2$ is the respiration rate at temperature $T2$ and $R1$ is the respiration rate at temperature $T1$ [20].

### 2.4. Weight Loss Rate and Quality

2.4.1. Weight Loss Rate

After harvest, samples were weighed using a digital balance (Shimadzu; UW4200H), and 150 g $\pm$ 10 g winged bean pods were used as a unit for weighing and recording. After precooling, the pods were stored for 3, 7, 10, and 14 days before weighing. The following formula was used to calculate the weight loss rate of winged bean pods:

$$\text{Weight loss rate } (\%) = \frac{W1 - W\alpha}{W1} * 100\%$$

where $W1$ is the initial weight after harvesting and $W\alpha$ is the weight after precooling and storage. Five replicates were used per treatment.

2.4.2. Quality

Peel Color Changes

One side of the winged bean pod was used for measurements. Two opposite points were measured and averaged. During measurement, a black cloth was used to cover the pods to avoid affecting the data. A colorimeter (Nippon Denshoku Industries Co., Ltd.; model ZE-2000) was used to measure the $L$, $a$, and $b$ values. Standard plates with Y = 3.97, X = 1.96, and Z = 10.41 were used for calibration. The greater the brightness (L), the brighter the sample, and the lower the L, that is, the darker the sample. The upper and lower limits were 0 and 100, respectively. The hue angle ($\theta$ value) was calculated as $|b*/a*|\tan^{-1}$ to indicate peel color changes. A hue angle of 0° indicates red, 90° indicates yellow, 180° indicates green, and 270° indicates blue. The chroma (C value) was calculated as $(a*^2 + b*^2)^{1/2}$. There higher the C value, the richer the chroma [21].

Texture

A force gauge (Shimpo, FGP-20) was used for analysis. The pod texture was analyzed using the cutting mode. A V-type probe was used to simulate the force required for teeth to chew pods, and the contact area thickness was 0.52 mm. The middle segment of each pod was used for measurement. Five pods were measured, and the values were averaged. The unit for texture was N.

Total Soluble Solids (TSS)

Using a handheld refractometer (Atago; N-1E), the TSS content, expressed as °Brix, of the winged bean juice was measured.

Ascorbic Acid

The method proposed by Nielsen [22] was modified and used for ascorbic acid analysis. Briefly, to 10 g of pods, 50 mL of 3% metaphosphoric acid ($HPO_3$) was added. The samples were evenly mixed and passed through filter paper. Next, 5 mL of the filtrate was collected, and 5 mL of metaphosphoric acid was added. Therefore, indophenol titration was performed until the solution turned pink. This was compared with the concentration of the standard in which 1 mm of ascorbic acid was used for titration. The calculation formula is as follows:

$$\text{Vitamin C (mg/100 g)} = \frac{S}{T} \times \frac{V_1}{V_2} \times \frac{1}{W} \times 100$$

where $S$ is the volume of indophenol used for sample titration (mL), $T$ is the volume of indophenol used for the titration of the ascorbic acid standard (mL), $V_1$ is the volume of metaphosphoric acid (50 mL), $V_2$ is the volume of filtrate (5 mL), and $W$ is the sample weight (g).

*2.5. Decay Rate and Storage Life*

A pod was considered to have decayed when mold, decay, and rotting were observed on the pod in the package. The decay rate was recorded based on the appearance of every bag of pods. The investigation was performed on days 3, 7, 10, and 14 of storage. The end of shelf life was defined as when 10% or more pods in the packaging had withered or yellowed, resulting in the loss of product value. Ten replicates were used for each treatment to determine the decay rate and shelf life.

*2.6. Statistical Analysis*

SAS 10.0 was used for the ANOVA. The least significant difference analysis (LSD) was applied with a 0.05 confidence interval ($p < 0.05$). SigmaPlot 10.0 was used to plot graphs.

## 3. Results

*3.1. Respiration Rate and Ethylene Production*

Temperature is an important factor that affects the crop's physiological and metabolic rate. Generally, the lower the temperature, the slower the respiration rate and the lower the ethylene yield. After harvest, the winged beans were stored at 5 °C, 10 °C, 15 °C, and 25 °C, and the mean respiration rate and ethylene yield of the pods were investigated after 3 days of storage. Both the pod respiration rate and ethylene yield decreased as the storage temperature decreased (Table 1). The respiration rates of the pods stored at 5 °C, 15 °C, and 25° were $10.60 \pm 4.16$, $25.70 \pm 5.47$, and $41.07 \pm 10.60$ mg $CO_2$ kg$^{-1}$·h$^{-1}$, respectively. $Q_{10}^R$ is a marker of respiration temperature sensitivity and indicates the relative increase in the respiration rate with every 10 °C increase in temperature. The $Q_{10}^R$ of the winged bean pods at 5 to 15 °C and 15 to 25 °C was 2.42 times and 1.59 times, respectively.

**Table 1.** The effects of different temperatures on the winged bean pod's respiration rate and ethylene production.

| Temperature (°C) | Respiration Rate (mg $CO_2$ kg$^{-1}$·h$^{-1}$) | Ethylene Production (µL $C_2H_4$ kg$^{-1}$·h$^{-1}$) |
|---|---|---|
| 5 | $10.60 \pm 4.16$ | $0.23 \pm 0.05$ |
| 10 | $18.81 \pm 3.17$ | $0.25 \pm 0.07$ |
| 15 | $25.70 \pm 5.47$ | $0.30 \pm 0.07$ |
| 25 | $41.07 \pm 10.60$ | $0.42 \pm 0.09$ |

Data are expressed as mean $\pm$ SD of three replicates ($n = 3$).

After harvesting, the ethylene production of the pods decreased as the storage temperature decreased. The ethylene production of the pods at a storage temperature of 5–15 °C was 0.23–0.30 µL $C_2H_4$ kg$^{-1}$·h$^{-1}$, and the ethylene production increased significantly to $0.42 \pm 0.09$ µL $C_2H_4$ kg$^{-1}$·h$^{-1}$ at the storage temperature of 25 °C.

*3.2. Cooling Process of Pods*

FC is suitable for many horticultural crop products. In the present study, a commercial small top-suction forced-air precooling machine was used for winged bean pod precooling, and around 16–20 kg pods were processed each time. The pod temperature was 25.3 °C when harvested on 29 November 2021 and sent for packaging. FC was used to rapidly remove field heat from winged bean pods and cool them. The changes in dimensionless temperature at the center of the pods in FC and RC were recorded (Figure 1). When FC was employed, the mean dimensionless temperature of 0.5 (1/2 precooling time) was 5.8 min, and the mean dimensionless temperature of 0.125 (7/8 precooling time) was 14.7 min. When RC was employed, the mean dimensionless temperature of 0.5 was 55.8 min, and the mean dimensionless temperature of 0.125 was 172 min. Using 1/2 and 7/8 of the precooling times as the bases, the cooling rate of FC was 9.6 times and 11.7 times that of RC, respectively.

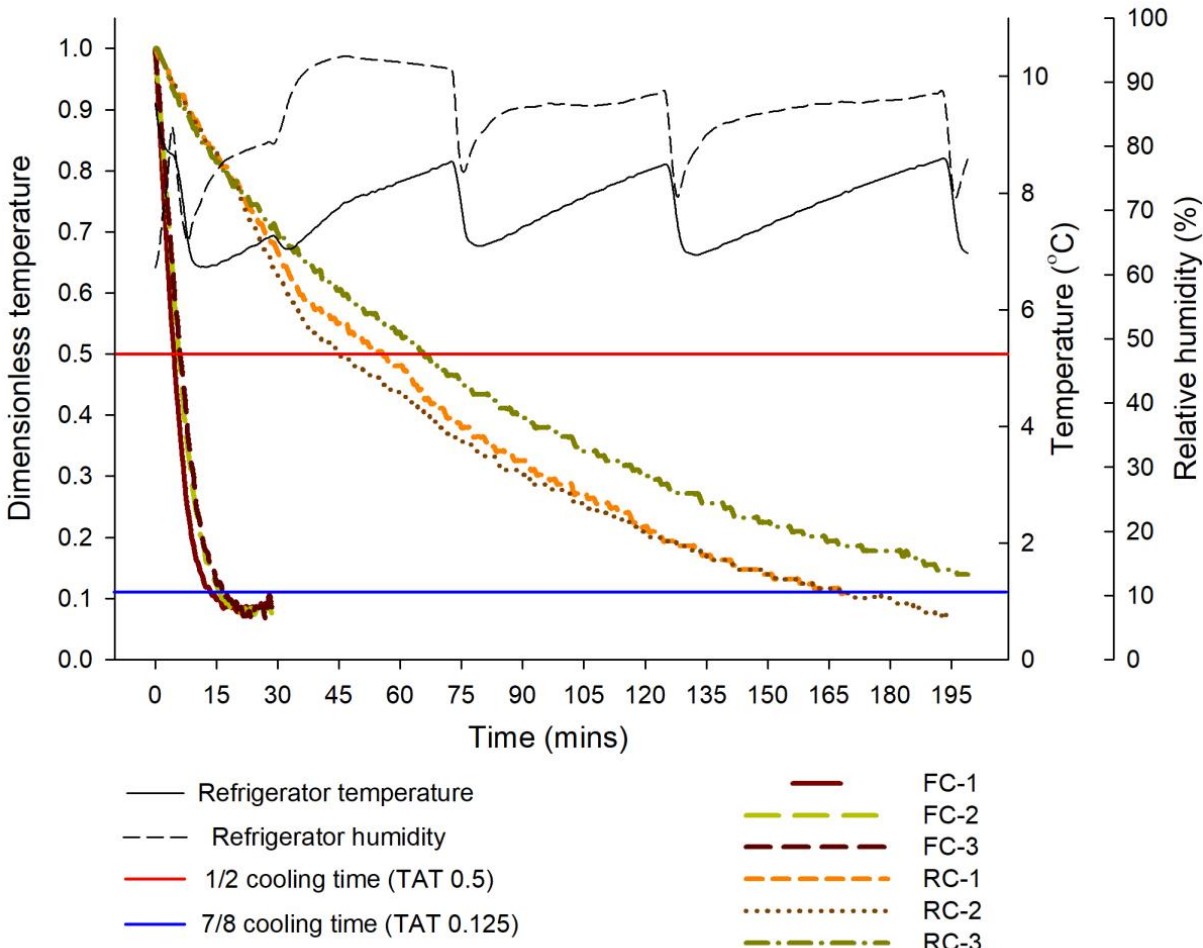

**Figure 1.** Changes in dimensionless temperature at the center of the pods in FC and RC groups. The red and blue lines are the 1/2 and 7/8 precooling times, respectively. The black solid and dotted lines indicate the cold storage temperature and RH, respectively.

### 3.3. Weight Loss Rate and Quality

FC and RC use cold air to remove field heat from the product, although this increases moisture loss. After FC was applied to remove field heat, the weight loss rate at 7/8 precooling time was 0.92%, significantly lower than that after RC was applied (1.98%) (Figure 2). Both FC and RC significantly increased the shelf life and weight loss rate during the storage period. At 3 and 14 days of storage, the weight loss rate of FC was significantly lower than that of RC. At 7 and 10 days of storage, the weight loss rate of FC was lower than that of RC, but there was a non-significant difference between the two groups in statistical analysis.

Table 2 shows the color changes of winged beans in different precooling groups during storage. On day 7 of storage, the color of winged beans treated using FC changed from green to yellow, and there were significant differences in color parameters compared with RC. However, after 14 days of storage, the hue angle decreased, and the color of all pods changed from green to yellow under both FC and RC treatments, with no significant differences in color parameters.

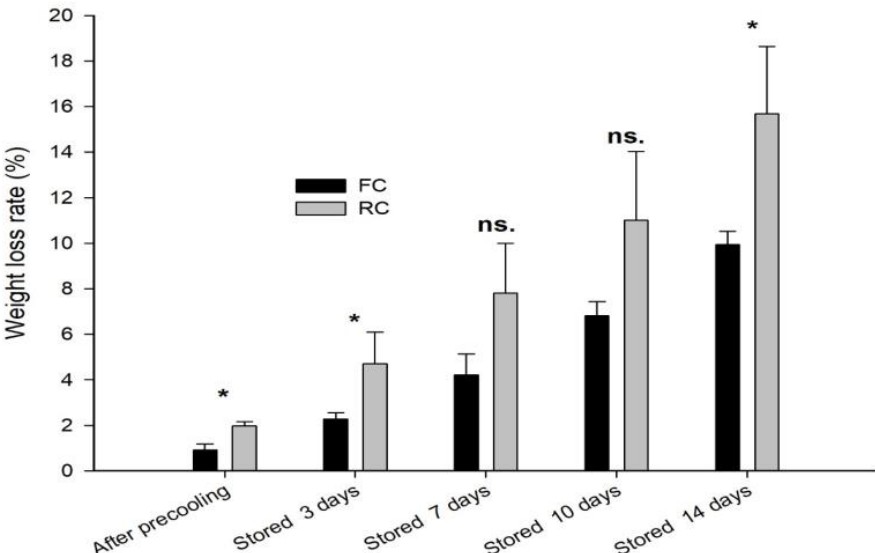

**Figure 2.** Effects of FC and RC on the weight loss rate during the storage of winged bean pods. * Significant difference between the two treatments according to the least significant difference test ($p < 0.05$). ns.—non-significant difference between the two groups. Data are expressed as mean ± SD of five replicates ($n = 5$).

**Table 2.** Effects of FC and RC on changes in pod color during the storage of winged bean pods.

| Storage at 12 °C | Treatment | Color | | |
| --- | --- | --- | --- | --- |
| | | *L* | Hue Angle (*h*) | Chroma (θ) |
| Before storage | | 48.6 ± 4.8 | 109.1 ± 3.7 | 39.6 ± 6.2 |
| 7 days | FC | 46.6 ± 6.7 b $^z$ | 105.1 ± 4.3 b | 34.9 ± 3.9 b |
| | RC | 49.7 ± 3.9 a | 111.0 ± 3.0 a | 39.8 ± 5.4 a |
| 14 days | FC | 43.2 ± 6.7 a | 102.8 ± 4.8 a | 36.3 ± 4.0 a |
| | RC | 43.4 ± 6.5 a | 102.5 ± 6.2 a | 36.7 ± 4.2 a |

$^z$ Data represent the mean ± SD of three replicates. Different letters within a column for the same day indicate significantly different values according to one-way ANOVA, followed by the least significant difference test ($p < 0.05$).

During the storage period, winged bean pod texture is an important factor affecting taste. Before storage, the texture was 49.5 N. During storage, the mean texture of the pods that underwent FC was higher than that of the pods that underwent RC. On day 7 of storage, the texture of the pods from the FC group (62.2 N) was significantly better than that of those in the RC group (48.2 N) (Figure 3A). The pod texture gradually deteriorated over storage time.

Furthermore, the TSS content of the pods before storage was 2.68°Brix, and the content gradually decreased over storage time (Figure 3B). On day 14 of storage, the TSS content of the RC group (2.13°Brix) was significantly higher than the FC group (1.85°Brix).

The ascorbic acid content of winged bean pods gradually decreased over storage time (Figure 3C). Before storage, the ascorbic acid content of the pods was 11.59 mg·100 g$^{-1}$ FW. On day 7 of storage, the ascorbic acid content of the pods in the FC group (10.8 mg·100 g$^{-1}$ FW) was significantly higher than that of the pods in the RC group (9.29 mg·100 g$^{-1}$ FW).

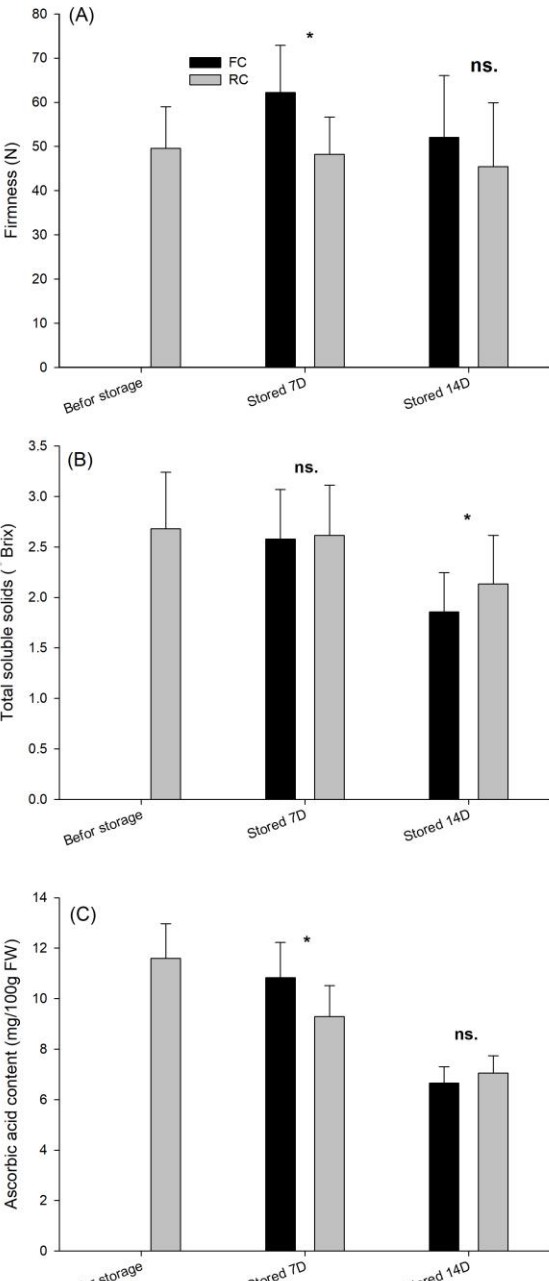

**Figure 3.** Effects of FC and RC on the (**A**) texture, (**B**) total soluble solids, and (**C**) ascorbic acid content of winged bean pods during storage. * Significant differences between the two treatments according to the least significant difference test ($p < 0.05$). ns.—non-significant difference between the two groups. Data are expressed as mean $\pm$ SD of three replicates ($n = 3$).

### 3.4. Decay Rate and Storage Life

Before storage, polyethylene bags were used for packaging the winged bean pods. As storage time increased, the wings of the pods started to wither and brown. Subsequently, the front, stems, and brown sites turned moldy and decayed. After storage at 12 °C for 14 days, the wings of the pods in the RC group showed significant and severe withering, browning, and decay (Figure 4A), whereas those in the FC group showed only slight browning (Figure 4B).

During the storage period, the front, stems, and brown sites of winged bean pods tend to develop microorganism-induced decay. Traces of decay (4%) occurred in winged beans in the RC group on day 7 of storage, and the decay rate increases with storage time. On

day 14 of storage, the decay rate was 52%. Meanwhile, FC could delay decay. On day 10 of storage, the decay rate was 4%, which increased to only 18% on day 14. During the storage period, the decay rate of the pods in the FC group was significantly lower than that of the pods in the RC group (Figure 5A).

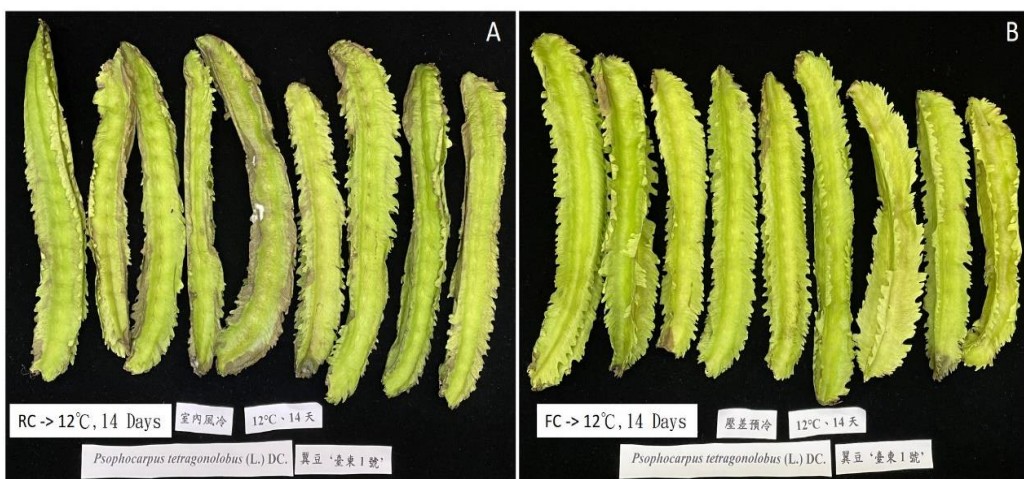

**Figure 4.** Pod appearance after FC and RC and storage at 12 °C for 14 days: (**A**) Room cooling. (**B**) Forced-air precooling.

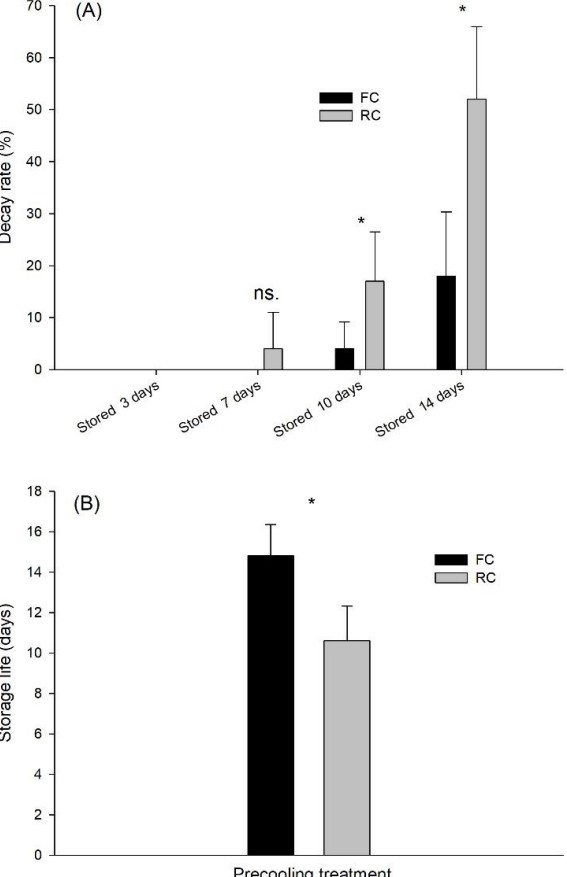

**Figure 5.** Effects of FC and RC on the (**A**) decay rate and (**B**) shelf life during the storage of winged bean pods. * Significant differences between the two treatments according to the least significant difference test ($p < 0.05$). ns.—non-significant difference between the two groups. Data are expressed as mean ± SD of ten replicates ($n = 10$).

When withering, browning, and decay occur in winged bean pods, the pods are considered to have lost their product value and shelf life. During 12 °C storage, the mean shelf life of winged bean pods in the FC group was 14.8 days, which was significantly longer than that of the pods in the RC group (10.6 days) (Figure 5B). Therefore, postharvest removal of field heat from winged beans using FC can significantly extend the shelf life of the pods.

## 4. Discussion

The postharvest preservation of winged beans is extremely crucial to ensure market acceptability. In this context, the harvest time and duration from harvest to cooling are critical. Often, the distance between the field and the cooling facility is long, which delays pod cooling, posing a challenge for producers. In addition, there are no packaging facilities in the winged bean fields of Taitung, and immediate cooling and removal of field heat are, therefore, impossible. This affects winged bean storage quality and shelf life. In the present study, a top-suction forced-air precooling machine was used to accelerate the postharvest cooling of winged beans, delay the degradation of the product, and extend the shelf life of the pods. Our results showed that when FC was used for postharvest removal of field heat from winged bean pods, the cooling rate of FC was significantly higher than that of RC. Simultaneously, FC decreased the weight loss rate of the pods after precooling and during storage. Furthermore, FC could maintain the winged bean pod quality during storage, decrease the pod decay rate, and effectively extend the shelf life of the pods.

Appropriate temperature control measures between harvesting and sale are the most effective methods for ensuring product quality [23]. As such, storing postharvest vegetables at a low temperature (20 °C) decreases physiological and metabolic activity. However, even a delay of 1 h decreases the shelf life by 1 day [24,25]. The respiration and metabolic processes of harvested crops are associated with the temperature of the surrounding environment [26,27]. The present study showed that winged bean pods' respiration rate and ethylene production decreased as storage temperature decreased. The $Q_{10}$ of winged bean pods at 5–15 °C and 5–25 °C was 2.5 times and 1.59 times, respectively, indicating that cooling is vital to winged bean's physiological and metabolic activities. Without a doubt, the most important external factor affecting breathing is temperature. This is because the temperature profoundly affects the rate of biological reactions, such as metabolism and respiration [28,29]. Within the physiological range of most crops, 0 to 30 °C (32 to 86 °F), elevated temperatures cause an exponential increase in respiration. Van 't Hoff Rule states that the physiological metabolism rate increases two to three times for every 10 °C increase over the sales temperature range [30].

In the present study, when FC was employed, the mean dimensionless temperature of 0.5 was 5.8 min, and the mean dimensionless temperature of 0.125 was 14.7 min. Meanwhile, when RC was employed, the mean dimensionless temperature of 0.5 was 55.8 min, and the mean dimensionless temperature of 0.125 was 172 min. Using 1/2 and 7/8 precooling times as the bases, the cooling rate of FC was 9.6 times and 11.7 times that of RC, respectively. The airflow rate of FC cooling is faster, which may cause the water loss rate per unit time to be higher than that of RC, but FC cooling significantly shortens the cooling time, so the weight loss of the pods treated with FC is lower than that of RC. Cortbaoui et al. [17] compared the cooling rates of FC and RC in sweet corn. At the wind speed of 3 L·s$^{-1}$·kg$^{-1}$, the 1/2 precooling time with RC was 436 min, whereas that with FC was only 47 min. Overall, the difference between the two treatments was nine times, similar to that in the present study. In another study, orange fruits of the same sizes were used to compare FC cooling rates at different wind speeds (0.2, 0.5, and 1 m·s$^{-1}$); as the wind speed increased, the 7/8 cooling time of the oranges became shorter [31].

Using different precooling methods (e.g., hydrocooling and vacuum precooling) for vegetables can help maintain product quality and prevent weight loss [32,33]. After FC, the weight loss rate of winged bean pods was significantly lower than that after RC. Likewise, during the storage period, the weight loss rate of FC-treated winged beans was lower

than that of RC-treated ones. In a study by Alibas and Koksal [34], the weight loss rate of cauliflower following FC and vacuum precooling was 2.89% and 4.55%, respectively. Therefore, the weight loss rate with FC was lower than with vacuum precooling. In another study, the fruit weight loss of rockmelon during postharvest storage was affected by the hydrocooling treatment and storage durations. Compared with that of fruits subjected to hydrocooling, the weight loss rate of fruits subjected to RC was higher, indicating that the RC-treated fruits exhibited a faster physiological metabolic rate, leading to more significant weight loss during the early stages of storage [35]. EO et al. [12] evaluated the effects of delayed postharvest cooling (0, 2, 4, 6, and 8 h postharvest delays) on French bean quality. Delayed pod cooling resulted in significant water and weight loss, while protein, lipid, and ash contents increased slightly. The authors recommended shortening the time between harvesting and cooling, which should not exceed 2 h to decrease the product degradation rate.

Postharvest vegetable quality decreases due to water loss. Moreover, agricultural products are prone to mechanical damage [36]. The texture of farm products is one of the important markers of quality, and the postharvest processing and transportation of fruits significantly affect their texture [37]. The activity of fruit softening-related enzymes, including polygalacturonase, β-galactosidase, pectin methylesterase, and pectin lyase, is temperature-dependent. Therefore, precooling and heat removal are indispensable, particularly for cherries, strawberries, and other farm products prone to rotting [38]. Sena et al. [39] studied the effects of hydrocooling on the postharvest quality of cashew apples and noted that hydrocooling delayed cashew apple softening, prevented weight loss, and decreased ascorbic acid content. In the present study, FC delayed texture deterioration during storage in winged beans. Hydrocooling could maintain cell turgor in rockmelon, thereby maintaining fruit cell wall structure and texture [35]. Choi et al. [40] evaluated the effects of storing 'Wonhwang' pears (*Pyrus pyrifolia* Nakai) at room temperature (20 ± 5 °C; 80–85% RH) for 3 weeks following precooling at 0 °C and 10 °C and storage at 23 °C for 24 h. The fruits were first precooled at 10 °C for 24 h, then stored at room temperature. After precooling, the fruits were firmer (fruit texture was 34.8 N after storage at room temperature for 21 days), the stone structure was intact, and the taste was good.

Many precooling-related studies have shown that rapid precooling of farm products can maintain quality. For instance, Zainal et al. [30] reported that hydrocooling and RC did not affect the peel brightness, chroma, and hue color changes during storage in rockmelons. Han et al. [41] explored the effects of precooling on the postharvest quality of black mulberry (*Morus nigra*) fruits. After precooling, fruits' TSS and titratable acid contents decreased gradually, but fruit color was maintained during storage. This may be associated with fruit respiration and metabolic rates. In the present study, FC delayed the textural degradation and increased the ascorbic acid content in winged beans. In a study by Rab et al. [42], immediate precooling of mushrooms after harvest resulted in the lowest decrease in fruit weight (9.167%) and the highest ascorbic acid content (8.06 mg·100 mL$^{-1}$). Furthermore, precooling significantly decreased the weight loss rate in tomatoes, delayed soluble solid accumulation, maintained the texture, and improved the ascorbic acid content of fruits compared with the control treatment.

The degree of deterioration of farm products after storage at 25 °C for 1 h postharvest was comparable to that after storage at 1 °C for 1 week, demonstrating the importance of precooling [43]. For instance, as the interval between harvesting and precooling increased, the degree of strawberry deterioration increased gradually. After storage at 30 °C for 2 h, only 80% of the strawberries could be shown, indicating that delay in precooling postharvest may lead to at least 10% of product loss [44]. In the present study, FC significantly decreased the decay rate of winged beans during storage and extended their shelf life. Cortbaoui et al. [17] reported that FC and RC extended the shelf life of sweet corn by up to 21 days when stored at 1 °C while maintaining high TSS and water content as well as good appearance. In litchis, hydrocooling could remove the field heat accumulated under the sun, decrease the physiological and metabolic rates, and prevent fruit rotting and peel

browning, ultimately extending the shelf life of fruits [45]. Rab et al. [42] reported that precooling could decrease the decay rate of tomatoes during storage to 15.76% (control 18.56%). These results indicate that rapid postharvest cooling of farm products is important for maintaining their quality and extending their shelf life.

## 5. Conclusions

Overall, rapid precooling of winged bean pods after harvesting is essential. The lower the postharvest pod temperature, the lower the respiration rate and ethylene production. The cooling rate of FC was 9.6 times faster than that of RC, and FC decreased the weight loss rate of the pods during storage. Further, FC could maintain winged bean pod quality during storage, decrease the pod decay rate, and effectively extend the shelf life. Therefore, FC is an effective precooling method to rapidly remove field heat postharvest and maintain the storage quality of winged beans.

**Author Contributions:** Conceptualization, Y.-S.L. and Y.-C.L.; methodology, M.-C.H. and Y.-S.L.; software, H.-Y.C.; validation, Y.-C.L., M.-C.H. and Z.-W.W.; formal analysis, H.-Y.C. and J.-Z.L.; investigation, Y.-S.L., J.-Z.L. and Z.-W.W.; resources, J.-Z.L.; data curation, Y.-C.L. and Y.-S.L.; writing—original draft preparation, Y.-C.L. and Y.-S.L.; writing—review and editing, Y.-S.L.; visualization, Y.-C.L.; supervision, H.-Y.C.; project administration, Y.-C.L. and Y.-S.L.; funding acquisition, Y.-S.L. All authors have read and agreed to the published version of the manuscript.

**Funding:** This research was funded by the Council of Agriculture, Executive Yuan, R.O.C.

**Conflicts of Interest:** The authors declare no conflict of interest.

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
