# Peer review of "Effects of Forced-Air Precooling on Postharvest Physiological and Storage Quality of Winged Beans"

_horticulturae, doi:10.3390/horticulturae9010045_

Round 1

Reviewer 1 Report

The manuscript about the effect of forced-air precooling on post-harvest physiological and storage quality of winged beans is relatively a simple work with practical application. However, the experiments look sound, and the manuscript is average written. Some issues that I point out are below. 

  • The abstract should be re-written since in the results section the authors should indicate the differences between treatments in percentage to be clearer for the readers.
  • The aim of this study is a weak point of the ms. I can’t see any tested hypothesis except a comparative study among the precooling methods.
  • I can’t understand the reason to present a and b values for color since hue and chroma are presented.
  • Please add the number of replicates in figure captions.

Reviewer 2 Report

A research study titled: “Effects of forced-air precooling on post-harvest physiological and storage quality of winged beans. This research relates to the Horticulturae’ journal’s scope in this special issue and focussed on effect of cooling techniques and quality of winged beans. However, my major consideration is results and discussion section. It should be improved for supporting previous cooling studies and cooling principles affecting the pod quality. Also, most discussion provided only each parameter but less discussion among or between quality parameters. Some results are not clear and lack of some points in Table 2 and figure 3. In this manuscript, this manuscript describes cooling technique for only field heat but cooling technique is method for both field heat and respiratory heat (respiration rate) removal. Thus, please recheck its definition and principle. Precooling was used throughout this manuscript and title. It should be used in the uniform word as cooling instead of precooling.

More detailed suggestions are provided below:

1. Title and abstract

L14- Please add 7/8 cooling time result.

L15- Please add storage condition.

2. Introduction

L42-45- Please give more details for pod deterioration such as wilting and browning.

L50- Please give more detail about respiration rate classification of winged bean (low, medium, High, very high) and compare it to other immature pods or other beans.

L59- What is cooling medium temperature level?

L60- It is general information. Please give more specific immature pods for both RC and FC.

L63- Total soluble solids

2. Materials and Methods

L82- Please give a range of pod size (g and length) before grading.

L87- relative humidity (RH). Please use RH instead of full word.

L 91- Please give more details for dimension, size and vent hole (%) of plastic basket, including stacking pattern.

L103- Please give more interval time for setting and data recording.

L 107- How many days?

L115- It should inform a closed system for respiration rate and ethylene production measurements,

L122- Please give a code of both column for CO2 and C2H4 measurements. Please add Q10 calculation and equation.

L131- Give a digital balancing and its model.

L140- Use italic style and *. L*, a*, b*

L148- Why do you use V-type probe? It should be cutting for bean testing. Please give a citation for this technique.

L176- t-test

3. Result and discussion

L190- Please add Q10 value in Table 1.

L207- Please report 1/2 and 7/8 times (min) for FC and RC.

L209- Figure 1 According to FC1-3 and RC1-3 lines are not clear. It should be report for average value in each FC or RC. Please make clearer lines in this graph.

L230- Please give more result for 3 days and 10 days.

L235- Weight loss of RC > FC. Why does texture value of FC > RC?

Figure 3 (B) Y axis (Total soluble solids) Where are the results of after precooling in both cooling?

Figure 4 This photo was a clear result. The quality of FC was better than RC with less browning incidence.  Why do color NS results in Table 2 for day14?

L299- Please discussion for respiration rate or Q10 in the previous studies. Also, what is effect of ethylene for the pod’s senescence.

L303- Please report 1/2 and 7/8 cooling time in each RC and FC for comparison. It could be easy to discuss with the previous studies.

L311-313- Remove it, please try to find more support information between FC and RC in immature pod. Why? High wind speed is not affecting weight loss of FC? FC should provide weight loss more than RC. Is has any information for supporting with other reason such as low speed, short time for cooling

L319-328- In this paragraph, please try to discuss why FC provided lower weight loss. You should discuss and compare FC with RC, not other cooling methods. It is not a clear understanding.

L335-336- Please reconsider about texture. According to texture change of immature pod, it should relate to water loss. Basically, a higher water loss increased firmness due to wilting symptom. In this study, a higher water loss found in RC but firmness is not difference. Please recheck a selected probe and firmness technique for texture measurement.

L350- In Figure 4, FC had a better pod quality than RC. Please recheck color results. It has any technical problems for measurement. How many pods for each treatment?

L363-377- You should emphasis your result in term of a significant decay reduction in FC. Have any decay differences between FC and RC after storage in other crops.

4. Conclusion

L381- It has no result from this study due to RR and ethylene production were not determined after two cooling methods.

Reviewer 3 Report

The authors have elucidated most of the research background, problem statement, and objectives well. The methodology is reproducible and results and discussion are adequately presented. However, a few syntax errors must be corrected in the manuscript.

All the comments are highlighted in the manuscript pdf file.

Thank you!
